# InfoCNF: Efficient Conditional Continuous Normalizing Flow Using Adaptive Solvers

## Abstract

Continuous Normalizing Flows (CNFs) have emerged as promising deep generative models for a wide range of tasks thanks to their invertibility and exact likelihood estimation. However, conditioning CNFs on signals of interest for conditional image generation and downstream predictive tasks is inefficient due to the high-dimensional latent code generated by the model, which needs to be of the same size as the input data. In this paper, we propose *InfoCNF*, an efficient conditional CNF that partitions the latent space into a class-specific supervised code and an unsupervised code that shared among all classes for efficient use of labeled information. Since the partitioning strategy (slightly) increases the number of function evaluations (NFEs), InfoCNF also employs gating networks to learn the error tolerances of its ordinary differential equation (ODE) solvers for better speed and performance. We show empirically that InfoCNF improves the test accuracy over the baseline while yielding comparable likelihood scores and reducing the NFEs on CIFAR10. Furthermore, applying the same partitioning strategy in InfoCNF on time-series data helps improve extrapolation performance. Additional details at `https://sites.google.com/view/infocnf-iclr/`

## 1 Introduction

Invertible models are attractive modelling choice in a range of downstream tasks that require accurate densities including anomaly detection (Bishop, 1994; Chandola et al., 2009) and model-based reinforcement learning (Polydoros & Nalpantidis, 2017). These models enable exact latent-variable inference and likelihood estimation. A popular class of invertible models is the flow-based generative models (Dinh et al., 2017; Rezende & Mohamed, 2015; Kingma & Dhariwal, 2018; Grathwohl et al., 2018) that employ a change of variables to transform a simple distribution into more complicated ones while preserving the invertibility and exact likelihood estimation. However, computing the likelihood in flow-based models is expensive and usually requires restrictive constraints on the architecture in order to reduce the cost of computation. Recently, (Chen et al., 2018) introduced a new type of invertible model, named the *Continuous Normalizing Flow* (CNF), which employs ordinary differential equations (ODEs) to transform between the latent variables and the data. The use of continuous-time transformations in CNF, instead of the discrete ones, together with efficient numerical methods such as the Hutchinson's trace estimator (Hutchinson, 1989), helps reduce the cost of determinant computation from $\mathcal{O}(d^3)$ to $\mathcal{O}(d)$, where $d$ is the latent dimension. This improvement opens up opportunities to scale up invertible models to complex tasks on larger datasets where invertibility and exact inference have advantages.

Until recently, CNF has mostly been trained using unlabeled data. In order to take full advantage of the available labeled data, a *conditioning method* for CNF – which models the conditional likelihood, as well as the posterior, of the data and the labels – is needed. Existing approaches for conditioning flow-based models can be utilized, but we find that these methods often do not work well on CNF. This drawback is because popular conditioning methods for flow-based models, such as in (Kingma & Dhariwal, 2018), make use of the latent code for conditioning and introduce independent parameters for different class categories. However, in CNF, for invertibility, the dimension of the latent code needs to be the same as the dimension of the input data and therefore is substantial, which results in many unnecessary parameters. These additional but redundant parameters increase the complexity of the model and hinder learning efficiency. Such *overparametrization* also has a negative impact on other flow-based generative models, as was pointed out by (Liu et al., 2019), but is especially bad in the case of CNF. This is because the ODE solvers in CNF are sensitive to the complexity of the

model, and the number of function evaluations that the ODE solvers request in a single forward pass (NFEs) increases significantly as the complexity of the model increases, thereby slowing down the training. This growing NFEs issue has been observed in unconditional CNF but to a much lesser extent (Grathwohl et al., 2018). It poses a unique challenge to scale up CNF and its conditioned variants for real-world tasks and data.

Our contributions in this paper are as follows:

**Contribution 1:** We propose a simple and efficient conditioning approach for CNF, namely *InfoCNF*. Our method shares the high-level intuition with the InfoGAN (Chen et al., 2016), thus the eponym. In InfoCNF, *we partition the latent code into two separate parts: class-specific supervised code and unsupervised code which is shared between different classes* (see Figure 1). We use the supervised code to condition the model on the given supervised signal while the unsupervised code captures other latent variations in the data since it is trained using all label categories. The supervised code is also used for classification, thereby reducing the size of the classifier and facilitating the learning. Splitting the latent code into unsupervised and supervised parts allows the model to separate the learning of the task-relevant features and the learning of other features that help fit the data. We later show that the cross-entropy loss used to train InfoCNF corresponds to the mutual information between the generated image and codes in InfoGAN, which encourages the model to learn disentangled representations.

**Contribution 2:** We explore the speed-up achievable in InfoCNF by tuning the error tolerances of the ODE solvers in the model. ODE solvers can guarantee that the estimated solution is within a given error tolerance of the true solution. Increasing this tolerance enhances the precision of the solution but results in more iterations by the solver, which leads to higher NFEs and longer training time. However, when training a neural network, it might not be necessary to achieve high-precision activation, i.e. the solution of the corresponding ODE, at each layer. Some noise in the activations can help improve the generalization and robustness of the network (Gulcehre et al., 2016; Bengio et al., 2013; Nair & Hinton, 2010; Wang et al., 2018a). With carefully selected error tolerances, InfoCNF can gain higher speed and better performance. However, the process of manually tuning the tolerances is time-consuming and requires a large amount of computational budget. To overcome this limitation, *we propose a new method to learn the error tolerances of the ODE solvers in InfoCNF from batches of input data*. This approach employs learnable gating networks such as the convolutional neural networks to compute good error tolerances for the ODE solvers.

**Contribution 3:** We study methods to improve the large-batch training of InfoCNF including tuning and learning the error tolerances of the ODE solvers, as well as increasing the learning rates.

We conduct experiments on CIFAR10 and show that InfoCNF equipped with gating networks outperforms the baseline Conditional Continuous Normalizing Flow (CCNF) in test error and NFEs in both small-batch and large-batch training. In small-batch training, InfoCNF improves the test error over the baseline by 12%, and reduces the NFEs by 16%. When trained with large batches, InfoCNF attains a reduction of 10% in test error while decreasing the NFEs by 11% compared to CCNF. InfoCNF also achieves a slightly better negative log-likelihood (NLL) score than the baseline in large-batch training, but attains a slightly worse NLL score in small-batch training.

In order to better understand the impact of the gating approach to learn the error tolerances, we compare InfoCNF with and without the gating networks. In small-batch training, InfoCNF with gating networks achieves similar classification and density estimation performance as the same model without the gating networks, but reduces the NFEs by more than 21%. When trained with large batches, gating networks help attain a reduction of 5% in test error and a small improvement in NLLs. We also confirm the benefits of our gating approach on unconditional CNF and observe that on CIFAR10 learning the error tolerances helps reduce the NFEs by 15% while preserving the NLL. Furthermore, we explore the potential benefit of the partitioning strategy for time-series data. In our experiments, when the latent code is partitioned in the baseline LatentODE (Rubanova et al., 2019), the model achieves better performance in curve extrapolation.

## 2    Efficient Conditioning via Partitioned Latent Code

We will begin by establishing our notation and provide a brief review of the flow-based generative models. Throughout this paper, we denote vectors with lower-case bold letters (e.g., $\mathbf{x}$) and scalars with lower-case and no bolding (e.g., $x$). We use $\mathbf{x}$, $y$, and $\theta$ to denote the input data/input features, the

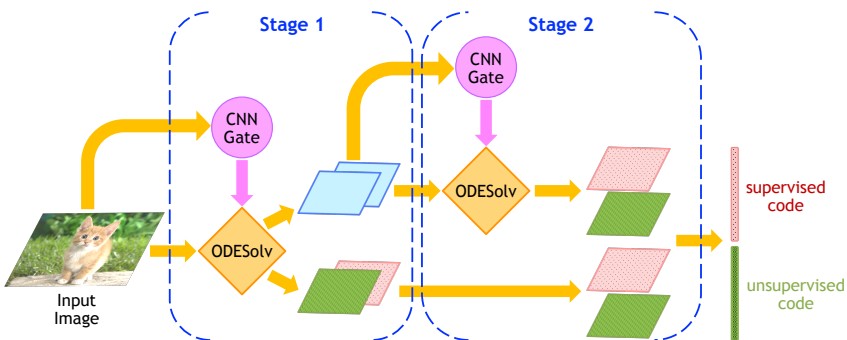

Figure 1: InfoCNF with CNN gates that learn the tolerances of the ODE solvers using reinforcement learning. The latent code in InfoCNF is split into the supervised and unsupervised codes. The supervised code is used for conditioning and classification. The unsupervised code captures other latent variations in the data.

supervised signals (e.g., the class labels), and the model's parameters respectively. The superscript $i$ is used to indicate the layer $i$. For example, $\mathbf{x}_i$ is the set of input features into layer $i$.

## 2.1 BACKGROUND ON FLOW-BASED GENERATIVE MODELS

Invertible flow-based generative models such as RealNVP (Dinh et al., 2017) and Glow (Kingma & Dhariwal, 2018) have drawn considerable interest recently. These models are composed of bijective transforms whose Jacobian matrices are invertible with tractable determinants. Let $f(\mathbf{z}; \theta)$ denote such a transform applied on the latent variable $\mathbf{z}$ to generate the data $\mathbf{x}$. Because of its bijective structure, the transform $f$ not only allows exact inference of $\mathbf{z}$ given $\mathbf{x}$ but also enables *exact* density evaluation via the change of variable formula:

$$\log p_x(\mathbf{x}) = \log p_z(\mathbf{z}) - \log \left| \det \left( \partial f(\mathbf{z}; \theta) / \partial \mathbf{z}^T \right) \right|. \tag{1}$$

The exact inference and exact density evaluation are preserved when stacking the bijective transforms into a deep generative model. The chain formed by the successive probability distributions generated by these transforms is called a normalizing flow, and the resulting generative models are called flow-based generative models. The distribution of the latent code $\mathbf{z}$ at the top of the model is usually chosen to be a factorized standard Gaussian to simplify the computation, and the parameters $\theta$ are learned by maximizing the exact log-likelihood $\log p(\mathbf{X}; \theta)$ where $\mathbf{X}$ is the training set containing all training data $\mathbf{x}$. While flow-based generative models enjoy nice properties, the requirements to ensure the invertibility and tractable computation restrict the expressive power of the model.

Recently, the Continuous Normalizing Flows (CNFs) have been explored in (Chen et al., 2018) to bypass the restrictive requirements in the flow-based generative models and allow the models to be expressive for more complicated tasks. CNF defines the invertible transforms via continuous-time dynamics. It models the latent variable $\mathbf{z}$ and the data $\mathbf{x}$ as values of a continuous-time variable $\mathbf{z}(t)$ at time $t_0$ and $t_1$, respectively. Given $\mathbf{z}$, CNF solves the initial value problem to find $\mathbf{x}$

$$\mathbf{z}(t_0) = \mathbf{z}, \, \partial \mathbf{z}(t) / \partial t = f(\mathbf{z}(t), t; \theta).$$

The change in log-density under this model follows the instantaneous change of variables formula

$$\log p(\mathbf{z}(t_1)) = \log p(\mathbf{z}(t_0)) - \int_{t_0}^{t_1} \mathrm{Tr} \left( \partial f(\mathbf{z}(t); \theta) / \partial \mathbf{z}(t) \right) dt. \tag{2}$$

Thus, CNF reduces the $\mathcal{O}(d^3)$ cost of computing the determinant to the $\mathcal{O}(d^2)$ cost of computing the trace. Taking advantage of the Hutchinson's trace estimator (Hutchinson, 1989), this computation cost can be reduced to $\mathcal{O}(d)$ (Grathwohl et al., 2018) where $d$ is the dimension of latent code $\mathbf{z}$.

## 2.2 CONDITIONING CNF VIA PARTITIONED LATENT CODE

**Conditional CNF:** To the best of our knowledge, there has not been a conditioning method particularly designed for CNF. However, since CNF belongs to the flow-based generative model family, conditioning methods for flow-based models can also be applied on CNF. In particular, the conditioning approach via a Gaussian Mixture Model (GMM) and an auxiliary classifier proposed in (Kingma & Dhariwal, 2018) has been widely used. We refer to CNF equipped with this type of conditioning as the Conditional Continuous Normalizing Flow (CCNF). We use CCNF as the baseline for comparison in our experiments. In CCNF, the distribution of the latent code $\mathbf{z}$ follows a GMM whose means

and scales are functions of the conditional signal $y$ and parameterized by simple neural networks. Furthermore, an additional predictive model is applied on $\mathbf{z}$ to model the distribution $p(y|\mathbf{z})$ via an auxiliary predictive task

$$\mathbf{z} \sim \mathcal{N}(\mathbf{z}|\boldsymbol{\mu}(y), \boldsymbol{\sigma}^2(y)); \ \boldsymbol{\mu}(y), \boldsymbol{\sigma}(y) = q_\phi(y); \ p(y|\mathbf{z}) = q_\theta(\mathbf{z}). \tag{3}$$

Here, $q_\phi(y)$ and $q_\theta(\mathbf{z})$ are usually chosen to be neural networks whose parameters are learned during training. While this conditioning method has been shown to enable flow-based models to do conditional image generation and perform predictive tasks such as object classification in (Kingma & Dhariwal, 2018), it is in fact rather inefficient. This is because the size of the latent code $\mathbf{z}$ in flow-based generative models is the same as the size of the input image and therefore often large. As a result, the conditioning network $q_\phi$ used to synthesize $\mathbf{z}$ and the predictive network $q_\theta$ applied on $\mathbf{z}$ introduce a significant amount of additional parameters for the model to learn.

**InfoCNF:** InfoCNF only uses a portion of the latent code $\mathbf{z}$ for conditioning. In particular, In-foCNF splits $\mathbf{z}$ into two non-overlapping parts – supervised latent code $\mathbf{z}_y$ and unsupervised latent code $\mathbf{z}_u$ – such that $\mathbf{z} = [\mathbf{z}_y, \mathbf{z}_u]$. The supervised latent code $\mathbf{z}_y$ captures the salient structured semantic features of the data distribution. In particular, denote the set of structured latent variables which account for those semantic features by $y_1, y_2, \cdots, y_L$. For simplicity and efficient computation, we assume that these latent variables follow a factored distribution such that $P(y_1, y_2, \cdots, y_L) = \prod_{i=1}^{L} P(y_i)$. The supervised code $\mathbf{z}_y$ is the concatenation of $\mathbf{z}_{y_1}, \mathbf{z}_{y_2}, \cdots, \mathbf{z}_{y_L}$ where $\mathbf{z}_{y_i}$ is the code that captures the latent variable $y_i$. We use $\mathbf{z}_y$ for conditioning the model. Similar to the conditional Glow, the distribution $p(\mathbf{z}_y)$ is modeled by a GMM , $\mathcal{N}(\mathbf{z}_y|\boldsymbol{\mu}(y), \boldsymbol{\sigma}^2(y))$, whose centers and scales are functions of the conditional signal $y$. As in Eq. (3), these functions are parameterized by a neural network $q_\phi(y)$. The posterior $p(y|\mathbf{z})$ is then approximated by another neural network $q_\theta(\mathbf{z}_y)$ applied on $\mathbf{z}_y$ to solve the corresponding predictive task. The unsupervised code $\mathbf{z}_u \sim \mathcal{N}(\mathbf{z}_u|0, I)$ can be considered as source of incompressible noise which accounts for other latent variations in the data.

We learn InfoCNF by optimizing the supervised loss from $q_\theta(\mathbf{z}_y)$ and the conditional log-likelihood $\log p(\mathbf{x}|y)$ of the model. The learning objective of InfoCNF is given by

$$\mathcal{J} = \mathcal{L}_{NLL}(\mathbf{x}|y) + \beta \mathcal{L}_{Xent}(\hat{y}, y), \tag{4}$$

where $\mathcal{L}_{Xent}(\hat{y}, y)$ is the cross-entropy loss between the estimated label $\hat{y}$ and the ground truth label $y$. $\beta$ is the weighting factor between the cross-entropy loss $\mathcal{L}_{Xent}(\hat{y}, y)$ and the conditional log-likelihood loss $\mathcal{L}_{NLL}(\mathbf{x}|y)$. $\mathcal{L}_{NLL}(\mathbf{x}|y)$ is given by

$$\mathcal{L}_{NLL}(\mathbf{x}|y) = \log p(\mathbf{x}|y) = \log p(\mathbf{z}_y|y) + \log p(\mathbf{z}_u) - \sum_{k=1}^{K} \int_0^1 \text{Tr}\left(\partial f_k(\mathbf{z}_k(t); \theta_k)/\partial \mathbf{z}_k(t)\right) dt, \tag{5}$$

where $k$ are indices for layers in the network and $\log p(\mathbf{z}_y|y), \log p(\mathbf{z}_u)$ are calculated from the formula for log-likelihood of a Gaussian distribution. In our notation, we set $\mathbf{z}_K = \mathbf{z}$ and $\mathbf{z}_0 = \mathbf{x}$. For each integral of the trace of the Jacobian in Eqn. 2, without generality, we choose $t_0 = 0$ and $t_1 = 1$.

**Connection to the mutual information in InfoGAN:** The mutual information between the generated images and the codes in InfoGAN is approximated by a variational lower bound via an "auxiliary" distribution, which is chosen to be a neural network. Since InfoCNF is an invertible model, the generated images from the model given the codes matches the input images. Thus, maximizing the mutual information between the generated images and the codes is equivalent to maximizing the cross-entropy loss between the estimated label and the ground truth label, which is the loss $\mathcal{L}_{Xent}(\hat{y}, y)$ in Eqn. 4. Thanks to the invertibility of InfoCNF, we can eliminate the need of using an additional "auxiliary" network.

Compared to CCNF, InfoCNF needs slightly fewer parameters since the size of the supervised code $\mathbf{z}_y$ is smaller than the size of $\mathbf{z}$. For example, in our experiments, $\mathbf{z}_y$ is only half the size of $\mathbf{z}$, and InfoCNF requires 4% less parameters than CCNF. This removal of unnecessary parameters helps facilitate the learning. As discussed in Section 4.3, our experiments on CIFAR10 suggest that InfoCNF requires significantly less NFEs from the ODE solvers than CCNF. This evidence indicates that the partition strategy in InfoCNF indeed helps alleviate the difficulty during the training and improves the learning of the model.

## 3 LEARNING ERROR TOLERANCES OF THE ODE SOLVERS

**Tuning the error tolerances:** We explore the possibility of improving InfoCNF by tuning the error tolerances of the ODE solvers in the model. The advantage of this approach is two-fold. First, it

reduces the number of function evaluations (NFEs) by the ODE solvers and, therefore, speeds up the training. Second, we hypothesize that the numerical errors from the solvers perturb the features and gradients, which provides additional regularization that helps improve the training of the model.

**Learning the error tolerances:** We extend our approach of tuning the error tolerances of the ODE solvers by allowing the model to learn those tolerances from the data. We propose InfoCNF with learned tolerances, which associates each ODE solver in InfoCNF with a gating network that computes the error tolerance of the solver such that the model achieves the best accuracy and negative log-likelihood with the minimal NFEs. These gates are learnable functions that map input data or features into the tolerance values. In our experiments, we use CNNs for the gates (see Figure 1).

The error tolerance decides how many iterations the solvers need to find the solution. This process is discrete and non-differentiable, which creates a unique challenge for training InfoCNF with learned tolerances. We exploit the reinforcement learning approach to solve this non-differentiable optimization and learn the parameters of the gating networks. In particular, at each gate, we formulate the task of learning the gating network as a policy optimization problem through reinforcement learning (Sutton & Barto, 1998) to find the optimal error tolerances.

In InfoCNF, we assume that the error tolerances can be modeled by a Gaussian distribution. The sample sequence of the error tolerances drawn from our policy starting with input $x$ is defined as $\mathbf{g} = [g_1, \cdots, g_N] \sim \pi_{F_\theta}$, where $F_\theta = [F_{1\theta}, \cdots, F_{N\theta}]$ is the sequence of network layers with parameters $\theta$ and $g_i \sim \mathcal{N}(g_i | \mu_i, \sigma_i^2)$ where $[\mu_i, \sigma_i^2] = G_i(\mathbf{x}_i)$ are the outputs of the gating network at layer $i$. Here, the policy is to decide which error tolerance to use and is defined as a function from input $\mathbf{x}_i$ to the probability distribution over the values of the error tolerances, $\pi(\mathbf{x}_i, i) = \mathbb{P}(G_i(\mathbf{x}_i = g_i))$. The CNN in the gating network is used to estimate the parameters of the probability distribution over the values of the error tolerances. We choose the rewards function $R_i = -\text{NFE}_i$, the negative of the number of function evaluations by the solver at layer $i$, so that our policy tries to fit the model well and do good classification while requiring less computation. The overall objective is given as:

$$\min_\theta \mathcal{J}(\theta) = \min \mathbb{E}_{\mathbf{x},y} \mathbb{E}_\mathbf{g} \left[ \mathcal{L}_{Xent}(\hat{y}, y) + \mathcal{L}_{NLL}(\mathbf{x}, y) - \frac{\alpha}{N} \sum_{i=1}^N R_i(\mathbf{g}) \right], \quad (6)$$

where the $\alpha$ balance between minimizing the prediction/NLL loss and maximizing the rewards.

Employing REINFORCE (Williams, 1992), we can derive the gradients $\nabla_\theta \mathcal{J}$. Defining $\pi_{F_\theta}(\mathbf{x}) = p_\theta(\mathbf{g}|\mathbf{x})$, $\mathcal{L} = \mathcal{L}_{Xent}(\hat{y}, y) + \mathcal{L}_{NLL}(\mathbf{x}, y)$ and $r_i = -[\mathcal{L} - \frac{\alpha}{N} \sum_{j=i}^N R_j]$, the gradients $\nabla_\theta \mathcal{J}$ is given by

$$\nabla_\theta \mathcal{J}(\theta) = \mathbb{E}_{\mathbf{x},y} \mathbb{E}_\mathbf{g} \nabla_\theta \mathcal{L} - \mathbb{E}_{\mathbf{x},y} \mathbb{E}_\mathbf{g} \sum_{i=1}^N \nabla_\theta \log p_\theta(g_i|\mathbf{x}) r_i \quad (7)$$

The first part of Eq. 7 is gradients of the cross-entropy and NLL loss while the second part corresponds to the REINFORCE gradients in which $r_i$ is the cumulative future rewards given the error tolerance estimated by the gating networks. This combination of supervised, unsupervised, and reinforcement learning encourages the model to achieve good performance on classification and density estimation while demanding less number of function evaluations by the ODE solvers.

## 4 EXPERIMENTAL RESULTS

In this section, we empirically demonstrate the advantage of InfoCNF over the baseline CCNF when trained on CIFAR10. Throughout the experiments, we equip InfoCNF with the gating networks to learn the error tolerances unless otherwise stated. Compared to CCNF, InfoCNF achieves significantly better test errors, smaller NFEs, and better (in large-batch training) or only slightly worse (in small-batch training) NLL. Furthermore, we observe that learning the error tolerances of the solvers helps improve InfoCNF in all criteria except for a slightly worse NLL in small-batch training and a similar NFEs in large-batch training. We also describe how we evaluate our model to make sure that reported results are not biased by the numerical error from the ODE solvers in section 4.2.

### 4.1 IMPLEMENTATION DETAILS

**Dataset:** We validate the advantages of our models on the CIFAR10 dataset. Uniform noise is added to the images, and during training, the data are randomly flipped horizontally.

**Networks:** We use the FFJORD multiscale architecture composed of multiple flows as in (Grathwohl et al., 2018) for our experiments. The network details can be found in Appendix B. When conditioning the model, we use separate 1-layer linear networks for the classifier $q_\theta$ and the condition encoder $q_\phi$. The parameters of the networks are initialized to zeros. In InfoCNF, we use half of the latent code for conditioning. We apply a dropout (Srivastava et al., 2014) of rate 0.5 on the linear classifier.

**Training:** We train both CCNF and InfoCNF with the Adam optimizer (Kingma & Ba, 2015). When using the batch size of 900, we train for 400 epochs with learning rate of 0.001 which was decayed to 0.0001 after 250 epochs. When using the batch size of 8,000, we train for 400 epochs with learning rate of 0.01 which was decayed to 0.001 at epoch 120.

## 4.2 EVALUATION PROCEDURE

The adaptive ODE solvers perform numerical integration and therefore have errors inherent in their outputs. When evaluating the models, we need to take these numerical errors into account. Since our method of learning the error tolerances is purely for training, a reasonable evaluation is to set the error tolerances to a small value at test time and report the results on the test set. In order to find which small value of the error tolerance to use for evaluation, we train InfoCNF on 1-D synthetic data. Since a valid probability density function needs to integrate to one, we take InfoCNF trained on the synthetic 1-D data sampled from a mixture of three Gaussians and compute the area under the curve using Riemann sum at different error tolerance values starting from the machine precision of $10^{-8}$ (see Appendix C for more details). Figure 2a shows that the numerical errors from InfoCNF is negligible when the tolerance is less than or equal to $10^{-5}$. Thus, in our experiments, we set tolerances to $10^{-5}$ at test time. In other to validate that a tolerance of $10^{-5}$ still yields negligible numerical errors on complex datasets like CIFAR10, we also evaluate our trained models using tolerances of $10^{-6}$, $10^{-7}$, and $10^{-8}$. We observe that when using those values for tolerances, our trained models yield the same test errors and NLLs as when using tolerance of $10^{-5}$. The NFEs of the ODE solvers reported in our experiments are computed by averaging the NFEs over training epochs. Figure 2b shows the learned error tolerances from InfoCNF trained on CIFAR10 using small batches of size 900. We validate that the learned tolerances from the train and test sets are similar and thus the learned tolerances from our model do not only overfit the training data. Evaluation using the learned error tolerances with different batch sizes is discussed in Appendix D.

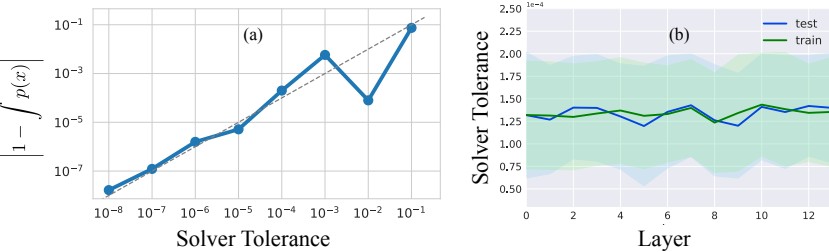

Figure 2: (a) Log plot that shows that the density under the model does integrate to one given sufficiently low tolerance. (b) Distribution of error tolerances learned by InfoCNF at each layer over CIFAR10 train and test sets.

## 4.3 INFOCNF VS. CCNF

Figure 3a shows that compared to CCNF, InfoCNF improves the test classification error on CIFAR10 by 12% when training with small batches. Interestingly, the advantage of InfoCNF over CCNF still holds when we train the models with large batch size of 8k (10% improvement in test classification error, see Figure 3d). While InfoCNF facilitates classification, it also attains similar NLLs compared to CCNF in small-batch training and better NLLs in large-batch training (Figure 3b, e). Figure 3c and f show the evolution of NFEs during training of InfoCNF and CCNF. In small-batch experiments, InfoCNF is much more efficient than CCNF in the first 240 epochs but then the NFEs of InfoCNF increases and exceeds the NFEs of CCNF. Overall, the NFEs from InfoCNF during training is 16% less than the NFEs from CCNF. When training with large batches, InfoCNF requires 11% less NFEs than CCNF.

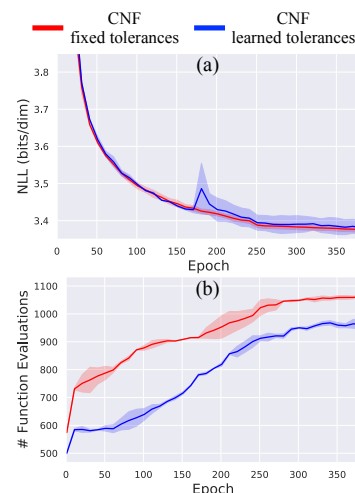

Figure 4: (a) Test NLLs, (b) NFEs of CNF with/without learned tolerances in small-batch training on CIFAR10.

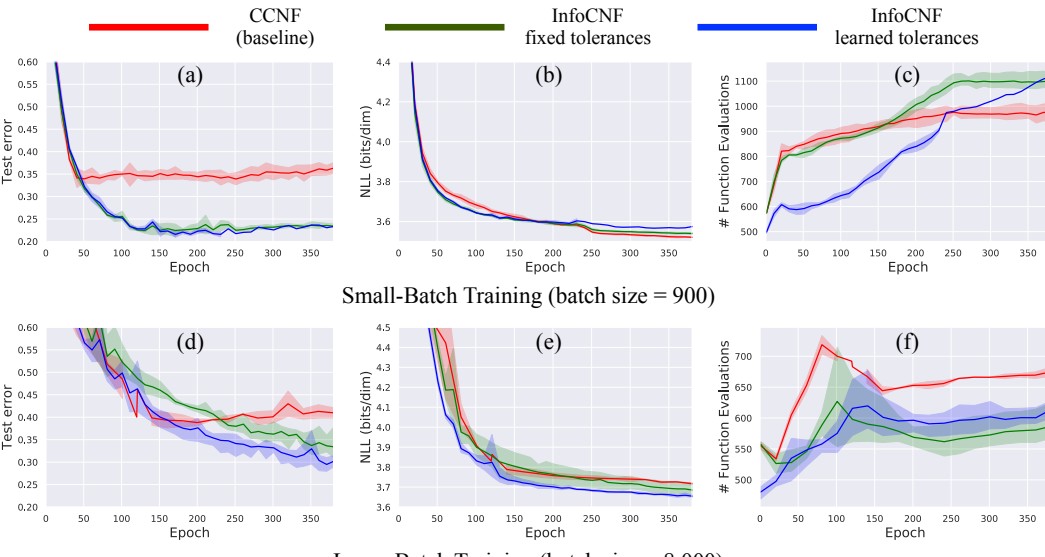

Figure 3: Evolution of test errors, test NLLs, and NFEs from the ODE solvers during the training of InfoCNF with learned tolerances vs. InfoCNF with fixed tolerances vs. CCNF (baseline) on CIFAR10 using small batch size (top) and large batch size (bottom). Each experiment is averaged over 3 runs.

## 4.4 IMPACTS OF GATING NETWORKS

We study the benefits of using gating networks to learn the error tolerances by comparing In-foCNF with learned error tolerances and with error tolerances set to $10^{-5}$. When training both models with large batches, InfoCNF with learned tolerances achieves the test error of 4% less than the test error attained by the same model with fixed error tolerances. InfoCNF with learned tolerances also yields slightly better NLLs (3.64 vs. 3.67). In small-batch training, both models achieve similar test errors and NLLs. Unlike with the test error and likelihood, the InfoCNF with learned tolerances significantly reduces the NFEs compared to InfoCNF with fixed error tolerances when both are trained with small batches–a reduction of 21%. In large-batch training, the NFEs from both models are similar. In summary, when InfoCNF with learned tolerances has the advantage over InfoCNF with tolerances set to $10^{-5}$, it is a notable improvement. Otherwise, InfoCNF with learned tolerances is as good as or only slightly worse than the its counterpart. We summarize the comparison between InfoCNF and CCNF in Table 1.

Table 1: Test errors, test NLLs, and NFEs of InfoCNF and CCNF.

| | Test error | NLLs | NFEs |
|---|---|---|---|
| | Small-batch | | |
| CCNF | $33.09 \pm 0.97$ | $\mathbf{3.511 \pm 0.005}$ | $924.12 \pm 22.64$ |
| InfoCNF with fixed tolerances | $21.50 \pm 0.29$ | $3.533 \pm 0.005$ | $984.92 \pm 10.34$ |
| InfoCNF with learned tolerances | $\mathbf{20.99 \pm 0.67}$ | $3.568 \pm 0.003$ | $\mathbf{775.98 \pm 56.73}$ |
| | Large-batch | | |
| CCNF | $38.14 \pm 1.14$ | $3.710 \pm 0.006$ | $660.71 \pm 4.70$ |
| InfoCNF with fixed tolerances | $31.64 \pm 3.63$ | $3.674 \pm 0.022$ | $\mathbf{582.22 \pm 10.16}$ |
| InfoCNF with learned tolerances | $\mathbf{27.88 \pm 0.73}$ | $\mathbf{3.638 \pm 0.006}$ | $589.69 \pm 11.20$ |

**Automatic Tuning vs. Manual Tuning:** In Figure 3d, e, and f, InfoCNF is trained with the manually-tuned ODE solvers' tolerances since otherwise the models are hard to train. Thus, for large-batch training, we are comparing the learned tolerances in InfoCNF with the manually tuned tolerances. Furthermore, Figure 10 in Appendix F shows similar comparison for small-batch training. In both experiments, we observe that *our automatic approach learns the tolerances which outperform the manually-tuned ones* in both classification and density estimation while being only slightly worse in term of NFEs. Also, *our automatic approach via reinforcement learning requires much less time and computational budget* to find the right values for the tolerances compared to the manual tuning.

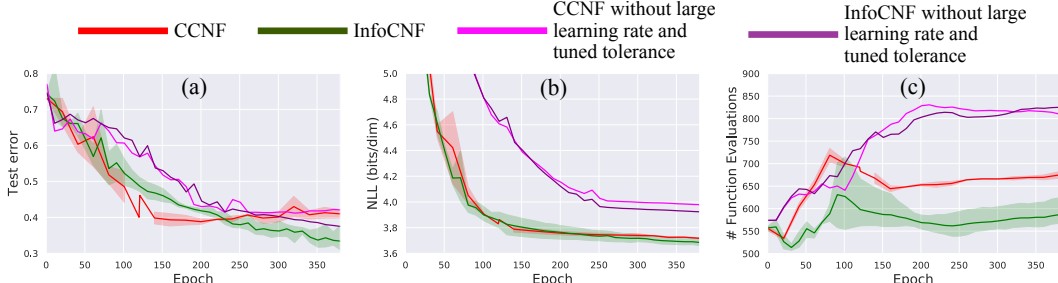

Figure 5: (a) Test errors, (b) test NLLs, and (c) NFEs of InfoCNF and CCNF trained on CIFAR10 using large batch size with and without large learning rate and manually-tuned error tolerance.

**Unconditional CNF:** Inspired by the success of InfoCNF, we explore whether the same reinforcement learning approach still work for unconditional models. We compare the baseline CNF in (Grathwohl et al., 2018) with CNF which uses the CNN gates to learn the error tolerance of the ODE solvers in small-batch training setting. While both models yield similar NLLs (3.37 bits/dim), CNF with learned tolerances significantly reduces the NFEs by 15% compared to the baseline CNF (see Figure 4).

### 4.5 IMPROVING LARGE-BATCH TRAINING OF INFOCNF AND CCNF

Training neural networks using large batches is much faster than small batches, but suffers from poor generalization (Keskar et al., 2017). In order to conduct meaningful experiments with large-batch training, we study methods to improve the performance of CCNF and InfoCNF. Our experiments confirm that tuning the error tolerance of the ODE solvers, together with using larger learning rate as suggested in (Keskar et al., 2017; Goyal et al., 2017), helps enhance the performance of both InfoCNF and CCNF, resulting in better test error, lower NLLs, and smaller NFEs (see Figure 5).

### 4.6 CONDITIONING VIA PARTITIONING ON TIME-SERIES DATA

We study the potential benefit of applying the conditioning method via partitioning the latent code in InfoCNF on time-series data. In this experiment, we choose the LatentODE (Chen et al., 2018) as the baseline model and conduct the experiment on the synthetic bi-directional spiral dataset based on the one proposed in (Chen et al., 2018). In particular, we first generate a fixed set of 5,000 2-dimensional spirals of length 1,000 as ground truth data: 2,500 curves are clockwise and the other 2,500 curves are counter-clockwise. These spirals have different

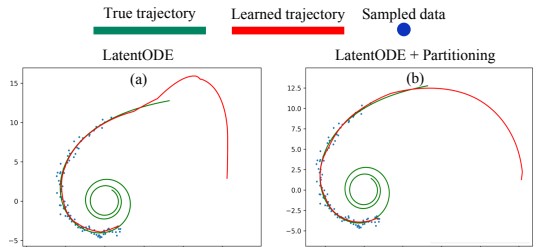

Figure 6: Trajectories learned by the LatentODE (a) without and (b) with the conditioning via partitioning the latent code.

parameters. We then randomly sample one spiral of 200 equally-spaced time steps from each of these ground truth spirals. We add Gaussian noise of mean 0 and standard deviation 0.3 to these small spirals to form the training set for our experiments. The test set is generated in the similar way. We train the LatentODE with and without our conditioning strategy for trajectory fitting and test the trained models for trajectory fitting and extrapolation on data in test set. More details on the dataset, network architecture, and training details are provided in Appendix E. We observe that the LatentODE equipped with our conditioning strategy outperforms the baseline Latent ODE on trajectory fitting and extrapolation tasks, especially in unseen domains.

## 5 RELATED WORK

**Conditional generative models:** (Nalisnick et al., 2019) applies a generalized linear model on the latent code of a flow-based generative model to compute both $p(\mathbf{x})$ and $p(y|\mathbf{x})$ exactly. Their method does not consider splitting the latent code and is complimentary to our partitioning approach. The conditioning approach proposed in (Ardizzone et al., 2019) is close to InfoCNF. However, in contrast to this method, InfoCNF does not penalize the mismatch between the joint distribution of the supervised and the unsupervised code $p(\mathbf{z}_y, \mathbf{z}_u)$ and the product of their marginal distributions $p(\mathbf{z}_y)p(\mathbf{z}_u)$. InfoCNF also does not minimize the MMD distance between the distribution of the backward prediction against the prior data distribution. Instead, we maximize the likelihood of the input image given the label $p(\mathbf{x}|y) = p(\mathbf{z}_y|y)p(\mathbf{z}_u)$. (Liu et al., 2019) uses an encoder to generate the

partitioned latent code for Glow. This encoder is trained by optimizing an adversarial loss as in the generative adversarial networks (GANs) (Goodfellow et al., 2014) so that its generated latent codes are indistinguishable from the latent codes computed by Glow for real data. Our InfoCNF directly splits the latent code without using an extra complicated architecture like GANs, which might introduce more instability into the ODE solvers and slow down the training.

**Adaptive computation:** The use of gating networks and reinforcement learning to learn a policy for adaptive computation has been studied to enhance the efficiency of the neural networks. (Wang et al., 2018b;c) employ gating networks to decide which blocks to skip in residual networks. (Liu et al., 2018) develops a reinforcement learning framework to automatically select compression techniques for a given DNN based on the usage demand. Other works including (Graves, 2016; Jernite et al., 2016; Figurnov et al., 2017; Chang et al., 2017) study methods to select the number of evaluations in recurrent and residual networks. To the best of our knowledge, our InfoCNF with learned tolerances is the first that learns the error tolerances of the ODE solvers in CNF.

**Large-batch training:** Various methods have been proposed to improve the generalization of neural networks trained with large batches by scaling the learning rate and scheduling the batch size. Among them are (Keskar et al., 2017; Goyal et al., 2017; Smith et al., 2018; You et al., 2017). These methods are complimentary to our approach of tuning the error tolerances.

## 6 CONCLUSIONS

We have developed an efficient framework, namely InfoCNF, for conditioning CNF via partitioning the latent code into the supervised and unsupervised part. We investigated the possibility of tuning the error tolerances of the ODE solvers to speed up and improve the performance of InfoCNF. We invented InfoCNF with gating networks that learns the error tolerances from the data. We empirically show the advantages of InfoCNF and InfoCNF with learned tolerances over the baseline CCNF. Finally, we study possibility of improving large-batch training of our models using large learning rates and learned error tolerances of the ODE solvers.

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

# APPENDIX

## A    GENERATED MNIST IMAGES FROM INFOCNF

We validate InfoCNF on the MNIST dataset. On MNIST, InfoCNF with learned error tolerances, InfoCNF with fixed error tolerances, and the baseline CCNF achieve similar NLLs and test errors. However, the InfoCNF with learned and fixed error tolerances are 1.5x and 1.04x faster than the baseline CCNF, respectively (416 NFEs/epoch vs. 589 NFEs/epoch vs. 611 NFEs/epochs). We include the detailed results in Table 2. All experiments are conducted with batch size 900, and the results are averaged over 3 runs.

Table 2: Test errors, test NLLs, and NFEs of InfoCNF and CCNF on MNIST.

|  | Test error | NLLs | NFEs |
|---|---|---|---|
|  |  | Small-batch |  |
| CCNF | $0.64 \pm 0.02$ | $\mathbf{1.016 \pm 0.003}$ | $611 \pm 7$ |
| InfoCNF with fixed tolerances | $\mathbf{0.60 \pm 0.01}$ | $1.030 \pm 0.004$ | $589 \pm 7$ |
| InfoCNF with learned tolerances | $0.61 \pm 0.02$ | $1.018 \pm 0.003$ | $\mathbf{416 \pm 5}$ |

Below is the MNIST images generated by InfoCNFwith learned error tolerances.

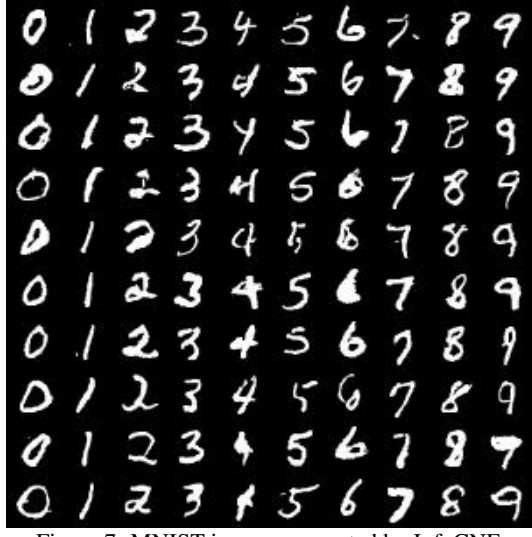

Figure 7: MNIST images generated by InfoCNF.

## B    NETWORK ARCHITECTURE FOR EXPERIMENTS ON CIFAR10

For experiments on CIFAR10, we use the FFJORD multiscale architecture composed of multiple flows as in (Grathwohl et al., 2018) for our experiments. The network uses 4 scale blocks, each of which contain 2 flows, a "squeeze" operator, and then other 2 flows. Each flow is made of 3 convolutional layers with 64 filters whose kernel size is 3. The squeeze operators are applied between flows to down-sample the spatial resolution of the images while increasing the number of channels. We apply the softplus nonlinearity at each layer. This architecture tries to reduce the dimensionality of the latent representation at each level while preserving invertibility. It was based on the multiscale architecture in (Dinh et al., 2017), which has been widely used as the base architecture for invertible models. Also, we parameterize $q_\theta$ and $q_\phi$ by separate linear networks.

## C    SYNTHETIC 1-D USED TO ESTIMATE THE VALUE OF ERROR TOLERANCES OF THE ODE SOLVERS AT TEST TIME

As mentioned in Section 4.2, in order to estimate the value of error tolerances of the ODE solvers which yields negligible numerical errors during test time, we train InfoCNF on a 1-D synthetic dataset and calculate the area under the curve at different error tolerance values starting from the machine precision of $10^{-8}$. Examples in this dataset are sampled from the mixture of Gaussians shown by

the blue curve in Figure 8. The orange curve in Figure 8 represents the distribution learned by our InfoCNF when the error tolerances are set to $10^{-5}$.

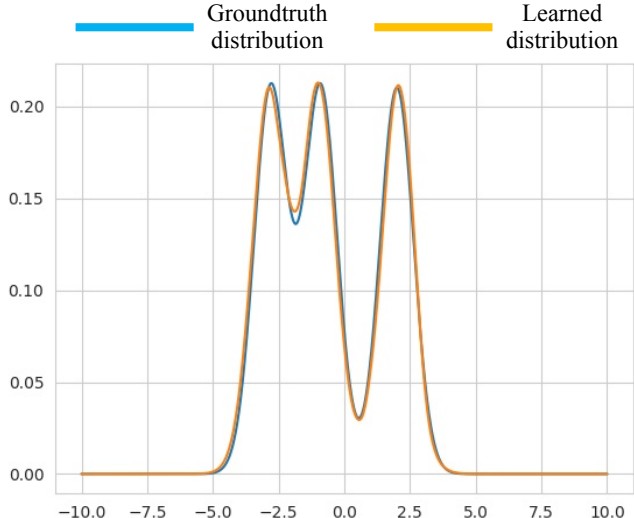

Figure 8: Distribution from which we sample the 1-D synthetic data for computing the value of error tolerances used to evaluate the trained InfoCNF (blue curve) and the distribution learned by InfoCNF (orange curve)

## D    EVALUATING THE TRAINED INFOCNF USING THE LEARNED ERROR TOLERANCES

We explore if the error tolerances computed from batches of input data can be used to evaluate the trained InfoCNF. First, we repeat the experiment on 1-D synthetic data described in Section 4.2 and Appendix C above using the learned error tolerances instead of the fixed error tolerances. We observe that the numerical error in this case is 0.00014, which is small enough. We further use the learned error tolerances to evaluate our trained InfoCNF on CIFAR10 with small-batches. Distribution of these error tolerances at different layers can be found in Figure 2b. The test error and NLL we obtain are $20.85 \pm 1.48$ and $3.566 \pm 0.003$, which are close enough to the results obtained when setting the error tolerances to $10^{-5}$ (test error and NLL in this case are $20.99 \pm 0.67$ and $3.568 \pm 0.003$, respectively, as shown in Table 1). Furthermore, when using the learned error tolerances to evaluate the trained model, we observe that the trained InfoCNF achieves similar classification errors, negative log-likelihoods (NLLs), and number of function evaluations (NFEs) with various small values for the batch size (e.g. 1, 500, 900, 1000, 2000). However, when we use large batch sizes for evaluation (e.g. 4000, 6000, 8000), those metrics get worse. This sensitivity to test batch size is because the error tolerances in InfoCNF are computed for each batch of input data.

## E    DATASET, NETWORK ARCHITECTURES, AND TRAINING DETAILS FOR EXPERIMENTS ON SYNTHETIC TIME-SERIES DATA

### E.1    DATASET

**Bi-directional Spiral Dataset of Varying Parameters:** In the experiments on time-series data in Section 4.6, we use the synthetic bi-directional spiral dataset based on the one proposed in (Chen et al., 2018). In particular, we first generate a fixed set of 5,000 2-dimensional spirals of length 1,000 as ground truth data: 2,500 curves are clockwise and the other 2,500 curves are counter-clockwise. These spirals have different parameters. The equations of the ground truth spirals are given below:

$$\text{Clockwise: } R = a + b \times \frac{50}{t}; \ x = R \times \cos(t) - 5; \ y = R \times \sin(t) \tag{8}$$

$$\text{Counter-Clockwise: } R = a + b \times t; \ x = R \times \cos(t) + 5; \ y = R \times \sin(t). \tag{9}$$

Here $a$ and $b$ serve as the system parameters and are sampled from the Gaussian distributions $\mathcal{N}(1.0, 0.08)$ and $\mathcal{N}(0.25, 0.03)$, respectively.

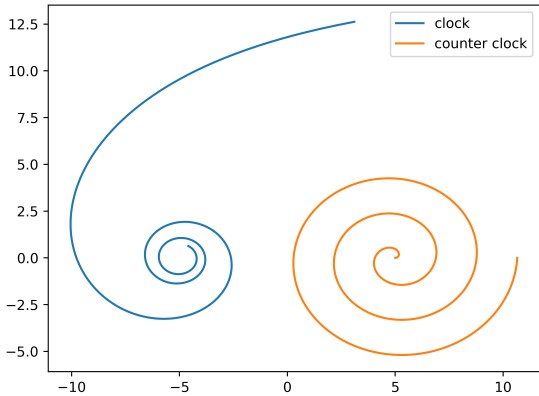

Figure 9: Ground truth spirals: One spiral is clockwise and another is counter-clockwise.

We then randomly sample one spiral of 200 equally-spaced time steps from each of these ground truth spirals. We add Gaussian noise of mean 0 and standard deviation 0.3 to these small spirals to form the training set for our experiments. The test set is generated in the similar way.

### E.2 NETWORK ARCHITECTURES

In our experiments, the baseline is the LatentODE model in (Chen et al., 2018). The RNN encoder that estimates the posterior $q_\phi(\mathbf{z}_{t_0}|\mathbf{x}_{t_0}, \mathbf{x}_{t_1}, \cdots, \mathbf{x}_{t_N})$ from the observations is fully connected and has 25 hidden units. The latent state $\mathbf{z}_{t_i}$ are 5-dimensional vectors. Different from the LatentODE, in our model, the first three dimensions of the initial latent state $\mathbf{z}_{t_0}$ are for the supervised code $\mathbf{z}_y$ and the other two dimensions are for the unsupervised code $\mathbf{z}_u$. Like the LatentODE, given the initial latent state $\mathbf{z}_{t_0}$, our model computes the future states $\mathbf{z}_{t_1}, \mathbf{z}_{t_2}, \cdots, \mathbf{z}_{t_N}$ by solving the equation $\partial \mathbf{z}(t)/\partial t = \boldsymbol{f}(\mathbf{z}(t), \boldsymbol{\theta}_f)$. The dynamic function $f$ is parameterized with a one-hidden-layer network of 20 hidden units. Also, the decoder that reconstructs $\hat{\mathbf{x}}_{t_i}$ from $\mathbf{z}_{t_i}$ is another one-hidden-layer network of 20 hidden units. Additionally, in our model, the conditioning function $q_\phi$ and the supervised function $q_\theta$ are linear networks. In our experiments, the conditioning/supervised signal $y$ are the values of the parameters $a$, $b$, and the direction of the spiral (i.e. clockwise or counter-clockwise).

### E.3 TRAINING DETAILS

We train the model with the Adam optimizer (Kingma & Ba, 2015). We use batch training with the learning rate of 0.001. The training is run for 20,000 epochs.

## F AUTOMATIC TUNING VS. MANUAL TUNING IN SMALL-BATCH TRAINING SETUP

We show the test error, NLLs, and the NFEs of InfoCNF with manually-tuned error tolerances for small-batch training on CIFAR10 (the yellow curves) in the Figure 10 in comparison with the results from CCNF, InfoCNF, and InfoCNF with learned tolerances in the main text. As can be seen, InfoCNF with learned tolerances, which learns the error tolerances, is still as good as InfoCNF with manually-tuned error tolerances (except for the slightly worse NFEs) when trained with small batches.

## G MARGINAL LOG-LIKELIHOOD RESULTS ON CIFAR10

The NLLs discussed in the main text are the conditional negative log-likelihoods $-\log p(\mathbf{x}|y)$. We would also like to compare the marginal negative log-likelihoods $-\log p(\mathbf{x})$ of CCNF, InfoCNF, and InfoCNF with learned tolerances. Figure 11 shows that like in the case of the conditional NLLs, InfoCNF with learned tolerances results in better marginal NLLs in large-batch training but slightly worse NLLs in small-batch training compared to InfoCNF and CCNF.

**Improving Large-Batch Training of InfoCNF and CCNF**: We would like to understand if using large learning rate and tuned error tolerances of the ODE solvers help improve the marginal NLLs in large-batch training. Figure 12 confirms that both InfoCNF and CCNF trained with large learning

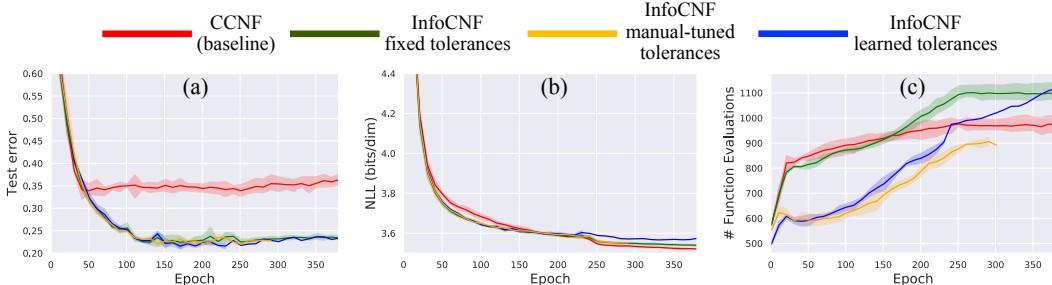

Figure 10: Evolution of test classification errors, test negative log-likelihoods, and the number of function evaluations in the ODE solvers during the training of InfoCNF with learned tolerances (blue) vs. InfoCNF with manually-tuned error tolerance (yellow) vs. InfoCNF with tolerances=$10^{-5}$ (green) vs. CCNF (red, baseline) on CIFAR10 using small batch size. Each experiment is averaged over 3 runs.

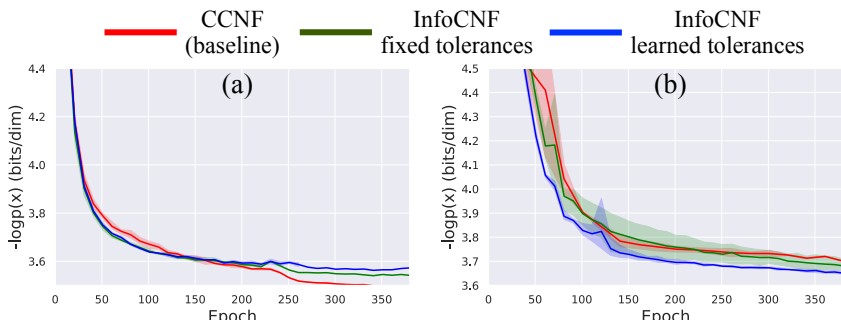

Figure 11: Evolution of test marginal negative log-likelihoods during the training of InfoCNF with learned tolerances (blue) vs. InfoCNF with fixed tolerances (green) vs. CCNF (red, baseline) on CIFAR10 using small batches of size 900 (left) and large batches of size 8,000 (right). Each experiment is averaged over 3 runs.

rate and tuned error tolerances yield much better marginal NLLs in large-batch training compared to the baseline training method which uses small learning rate and constant error tolerances.

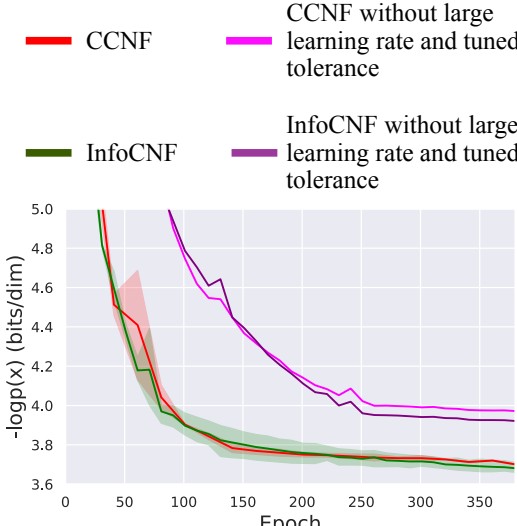

Figure 12: Test marginal NLLs of InfoCNF and CCNF trained on CIFAR10 using large batch size with and without large learning rate and manually-tuned error tolerance.

# H   LARGE-BATCH TRAINING VS. SMALL-BATCH TRAINING

Given promising results from training our models with large batches, we would like to compare large-batch with small-batch training in term of speed. Currently, the CNF-based models studied in our paper yield better test classification errors and negative log-likelihoods on CIFAR10 when trained

with small batches than with large batches. However, since large-batch training attains smaller NFEs than small-batch training, we would like to explore if the model trained with large batches can reach certain test classification errors and NLLs faster than the same model trained with small batches. In our experiments, we choose to study InfoCNF with learned tolerances since it yields the best test error and NLLs in large-batch training while requiring small NFEs. We compare InfoCNF with learned tolerances trained with large batches and with small batches. We also compare with the baseline CCNF trained with small batches. Figure 13 shows that InfoCNF with learned tolerances trained with large batches achieves better test error than CCNF trained with small batches while converging faster. However, it still lags behind CCNF trained with small batches in term of NLLs and InfoCNF with learned tolerances trained with small batches in both test error and NLLs. This suggests future work for improving large-batch training with the CNF-based models.

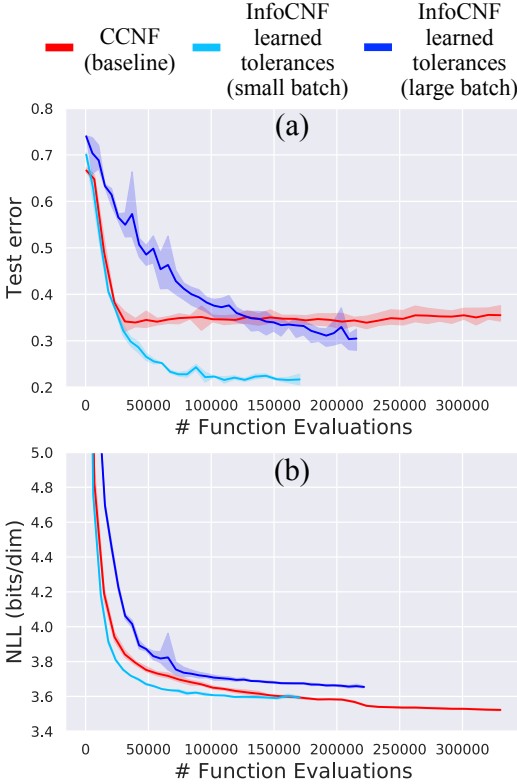

Figure 13: (a) Test error and (b) NLLs vs. NFEs during training of CNF and InfoCNF (with learned tolerances) on CIFAR10.

## I    INFOCNF WITH LEARNED TOLERANCES FACILITATES TRAINING OF LARGER MODELS

Since InfoCNF with learned tolerances reduces the NFEs while improving the test error and NLL, we would like to explore the possibility of using InfoCNF with learned tolerances for training larger models to gain better performance in classification and density estimation. We train a InfoCNF with learned tolerances with 4 flows per scale block (compared to 2 flows per scale block as in the original model) on CIFAR10 using large batches and compare the results with the original InfoCNF with learned tolerances and the baseline CCNF. Here we follow the same notation as in (Grathwohl et al., 2018) to describe the models. We call the large InfoCNF with learned tolerances with 4 flows per scale block the 2x-InfoCNF with learned tolerances. Figure 14 shows that the 2x-InfoCNF with learned tolerances yields better test errors and NLLs while increasing the NFEs compared to both InfoCNF and CCNF.

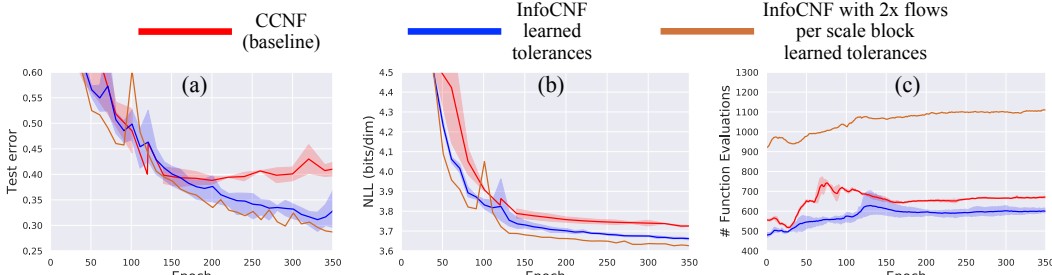

Figure 14: Evolution of test classification errors, test negative log-likelihoods, and the number of function evaluations in the ODE solvers during the training of the large size InfoCNF with 2x more flows per scale block and learned tolerances (brown) vs. InfoCNF with learned tolerances (blue) vs. CCNF (red, baseline) on CIFAR10 using large batchs of size 8,000.

