# OpenReview forum: "InfoCNF: Efficient Conditional Continuous Normalizing Flow Using Adaptive Solvers"
_ICLR.cc/2020/Conference — Reject_

### Official Review · AnonReviewer1 · 2019-10-23
**Official Blind Review #1**

**Rating:** 3

**Review:**

This paper examines the problem of extending continuous normalizing flows (CNFs) to conditional modeling. The authors propose a model, InfoCNF, which models a latent code split into two partitions: one that is unique to a class, and another that is more general. InfoCNF relies on the accuracy of ODE solvers, so the paper also proposes a method that learns optimal error tolerances of these solvers. They perform experiments on CIFAR10, showing that InfoCNF outperforms a baseline in accuracy and negative log-likelihood. They also ablate through experiments that demonstrate the utility of learning the ODE error tolerances.

CNFs are an exciting tool, and it's an important problem to devise methodology that applies CNFs to conditional modeling tasks. The idea of splitting latent codes into two separate components -- one supervised, the other more general -- is interesting. And the approach to learning error tolerances is a good idea.

The main drawback of this paper is the lack of clarity. It is poorly written and the presented model is not clearly motivated. Even after reading through the paper multiple times, I find it difficult to understand various aspects of InfoCNF. Below are some examples of this lack of clarity:

- When motivating Conditional CNF (CCNF), the details for training the model are unclear. What is the loss function, and how does it balance between modeling x and learning the auxiliary distribution? Although these may seem like small details, since InfoCNF builds off on CCNF, it is crucial to solidify an understanding of the training procedure and how the auxiliary task relates to modeling x. Moreover, the cited reference (Kingma and Dhariwal, 2018) does not contain these details (and there is no mention of object classification in the paper, contrary to the claim on page 3). It would be helpful to cite other references when mentioning that this approach is widely used.

- The definition of InfoCNF is unclear. The variable y has been used to denote the image label, so why are there now L latent variables y_1, ..., y_L? The following terms of equation 4 are undefined: L_{NLL}, L_{Xent}, and y_hat. Although some readers would be able to understand these definitions from context (flow-based NLL, cross entropy loss, and the logits provided by the auxiliary distribution q), they are never explicitly defined and result in confusion and ambiguity. Most importantly, p(x|z,y) is never made explicit; although one can infer from context that  is transformed to x using CNFs, it is never made explicit in the definition (and that these flows only appear in L_{NLL}). Overall, the motivation for splitting the latent code into two pieces is not clearly explained, and the paper should spend more time arguing for this.

The paper compares InfoCNF to a single baseline (CCNF). I understand that the paper is proposing the first method that combines CNFs with conditional modeling, but there are plenty of non-CNF flow-based conditional approaches that could've been compared. The paper goes over these approaches in Section 5 and discusses their differences with InfoCNF, but these are never experimented with. It seems possible that these models could be extended in a straightforward manner to use CNFs instead of other normalizing flows. Even comparing with these models without CNFs would have been interesting. I think that using a single baseline, instead of applying a more complete set of comparisons, hurts the persuasiveness of their method.

In addition to comparing to a single baseline, the paper only compares for a single dataset (CIFAR10). A more convincing experiments section would compare against more models on more than a single dataset.

The paper proposes a promising idea to an important problem. Due to the lack of clarity throughout the paper and incomplete comparisons, I would argue to (weakly) reject.

**Experience Assessment:**

I have published one or two papers in this area.

**Review Assessment: Checking Correctness Of Derivations And Theory:**

N/A

**Review Assessment: Checking Correctness Of Experiments:**

I assessed the sensibility of the experiments.

**Review Assessment: Thoroughness In Paper Reading:**

I read the paper thoroughly.

---

> ### Author Response · Authors · 2019-11-14
> **Motivation behind Splitting the Latent Code into Two Pieces**
>
> There are two motivations for splitting the latent code. First, we would like to reduce the size of the latent code used for classification. Since invertible models like InfoCNF require the latent code to be of the same size as the input images, this requirement significantly increases the size of the classifier, which in turn hinders the learning and reduces the classification accuracy. Second, splitting the latent code into unsupervised and supervised parts allows the model to separate the learning of important features for classification (and more general, the task-relevant features) and the learning of other features that help fit the data.  We have updated the introduction of our paper, especially Contribution 1, to address these two motivations.
>
> In addition, please refer to my reply to Reviewer #3: "InfoCNF Preserves the Beauty of InfoGAN " for the connection between the cross-entropy loss used to train InfoCNF and the mutual information between the generated image and codes in InfoGAN, which encourages the model to learn disentangled representations.

---

> ### Author Response · Authors · 2019-11-14
> **Comparison with Other Flow-based Conditional Approaches**
>
> While there have been other conditional approaches for flow-based models, they are either unnecessary complicated [1] or complementary to our approach [2]. It is also worth noting that the complicated approach using GAN in [1] will have a hard time to be used with the Continuous Normalizing Flow models since it will be likely to increase the number of function evaluations of the solvers by a great amount, thereby making the training more unstable and slower.
>
> References:
>
> [1] Rui Liu, Yu Liu, Xinyu Gong, Xiaogang Wang, and Hongsheng Li. Conditional adversarial generative flow for controllable image synthesis. arXiv preprint arXiv:1904.01782, 2019.
>
> [2] Eric Nalisnick, Akihiro Matsukawa, Yee Whye Teh, Dilan Gorur, and Balaji Lakshminarayanan. Hybrid models with deep and invertible features. In International Conference on Machine Learning, 2019.

---

> ### Author Response · Authors · 2019-11-14
> **Replies to Other Comments from Reviewer #1**
>
> 1) Improve the Clarity of the Paper
>
> We have updated our paper to provide definitions for L_xent, L_NLL,  p(x|z,y), and y_hat in equation (4) and (5) at the end of section 2.2. The \beta coefficient in equation (4) balances between modeling input image x and learning the task-relevant features. The training procedure in InfoCNF is as follows:
>
> a) Send a mini-batch of training images through the inverse model of InfoCNF to compute the supervised code z_y and the unsupervised code z_u. In this inference step, InfoCNF also computes the integrals in equation (5).
>
> b) The supervised code z_y is then used to calculate the cross-entropy loss L_xent(y_hat, y) between the estimated label y_hat and the ground truth label y in equation (4).
>
> c) The supervised code z_y and the unsupervised code z_u are used to compute the log-likelihoods log(p(z_y|y)) and log(p(z_u)). These two log-likelihoods and the integrals computed in the inference step are combined to calculate the negative log-likelihood loss L_NLL(x|y) according to equation (5). Note that the forward model of  InfoCNF is used to generate images and does not involve in the training. The forward and inverse model of InfoCNF share weights and are invertible to each other.
>
> d) Update the parameters of the model to maximize the total loss J = L_NLL + \beta * L_xent as specified in equation (4).
>
> e) Repeat step a, b, c, d above for the next mini-batch.
>
> Furthermore, y_1, …, y_L are different values for labels corresponding to class 1, .., L.
>
> In addition, please refer to our reply to Reviewer #3: “Replies to Other Comments from Review #3” for more details on the definition of error tolerances and problem statement for tuning these tolerances.
>
> 2) The Github code of the cited reference (Kingma and Dhariwal, 2018) contains details on conditioning their flow-based model
>
> Even though (Kingma and Dhariwal, 2018) paper does not contain these details, the usage of reparameterization via GMM approach for conditioning flow-based model can be found in their Github code. In particular, the subroutine  "def prior" in their model.py file implements this reparameterization strategy. Here is the link to their code:
>
> https://github.com/openai/glow/blob/master/model.py

---

> ### Author Response · Authors · 2019-11-15
> **Experimental Results on MNIST and Time-Series Data**
>
> Due to the lack of computational resources, we could not have experimental results on image benchmarks with higher resolutions like the ImageNet dataset. In fact, almost all papers on the Continuous Normalizing Flow and Neural ODEs can only provide experimental results on CIFAR10, MNIST and synthetic data.
>
> We have run additional experiments on the MNIST dataset. On MNIST, InfoCNF with learned error tolerances, InfoCNF with fixed error tolerances, and the baseline CCNF achieve similar NLLs and test errors. However, the InfoCNF with learned and fixed error tolerances are 1.5x and 1.04x faster than the baseline CCNF, respectively (416 NFEs/epoch vs. 589 NFEs/epoch vs. 611 NFEs/epochs). All experiments are conducted with batch size 900, and the results are averaged over 3 runs. We include the detailed results below. We have updated our paper to include these results with error bars in Section A of the Appendix. The MNIST images generated by InfoCNF with learned error tolerances are provided in the same section.
>
> 					                                  Test Error	        NLLs	 NFEs
> CCNF:                                                    	                0.64   		1.016   	   611
> InfoCNF without learning error tolerance:	0.60  		1.03  	   589
> InfoCNF with learning error tolerance:	        0.61		        1.018	   416
>
> Furthermore, Section 4.6: Conditioning via Partitioning on Time-Series Data in our paper studies the advantage of the partitioning strategy when applied on time-series data. We show that the baseline model equipped with our partitioning strategy achieves better performance on trajectory fitting and extrapolation tasks, especially in unseen domain. The experiments are conducted on the modified and more challenging version of the bi-directional spiral dataset proposed in [1], and the baseline model is chosen to be the Latent ODE [1]. More details on the dataset, network architecture, and training can be found in Append E of our paper.
>
> [1] Yulia Rubanova, Ricky TQ Chen, and David Duvenaud. Latent odes for irregularly-sampled time series. arXiv preprint arXiv:1907.03907, 2019

---

### Official Review · AnonReviewer2 · 2019-10-24
**Official Blind Review #2**

**Rating:** 6

**Review:**

This paper proposed a conditional CNF based on a similar intuition of the InfoGAN that partitions the latent space into a class-specific supervised code and an unsupervised code shared among all classes. To improve speed, the paper further proposed to employ gating networks to learn the error tolerance of its ODE solver. The experiments are performed on the CIFAR-10 dataset and synthetic time-series data.

The paper has addressed an important issue of investigating efficient conditional CNF. The general idea of the paper is clear, but I found certain parts can be improved, such as the formulation of InfoCNF. It seems the authors assume readers know InfoGAN well enough, which might not be the case.

My main concern is the limited evaluation as all the experiments are performed on the CIFAR-10 and synthetic data. Since the paper address efficient conditional CNF, it would make the claim much stronger if more experiments could be performed on larger images: if not the original imagenet, maybe imagenet-64 or imagenet-128 or image benchmarks with higher resolutions.

Why does InfoCNF achieve slightly worse NLL in small batch training, while it outperforms CCNF in all the other metrics? Do you have any explanations?


**Experience Assessment:**

I do not know much about this area.

**Review Assessment: Checking Correctness Of Derivations And Theory:**

I assessed the sensibility of the derivations and theory.

**Review Assessment: Checking Correctness Of Experiments:**

I assessed the sensibility of the experiments.

**Review Assessment: Thoroughness In Paper Reading:**

I read the paper at least twice and used my best judgement in assessing the paper.

---

> ### Author Response · Authors · 2019-11-14
> **Replies to Comments from Reviewer #2**
>
> 1) Formulation of InfoCNF
>
> Please refer to our replies to Reviewer #3 and #1: "Replies to Other Comments from Review #3" (part 2), "InfoCNF Preserves the Beauty of InfoGAN", "Replies to Other Comments from Reviewer #1" (part 1), and "Motivation behind Splitting the Latent Code into Two Pieces".
>
> 2) Why does InfoCNF Achieve Slightly Worse NLL in Small Batch Training, while It Outperforms CCNF in All the Other Metrics?
>
> Please refer to our reply to Reviewer #4: "Replies to Other Comments from Reviewer #4" (part 4).
>
> 3) Experimental Results on Other Datasets
>
> Please refer to our reply to Reviewer #1: "Experimental Results on MNIST and Time-Series Data".

---

### Official Review · AnonReviewer4 · 2019-11-04
**Official Blind Review #4**

**Rating:** 3

**Review:**

This paper suggests a methodology of partitioning latent code to a set of class-specific codes, to solve the inefficiency from the large size of the latent code in conditional normalizing flow. This inefficiency problem has been a weak point in the existing conditional normalizing flow models. Also, this work addresses the increase of the number of function evaluations which is caused by code partition, by optimizing the error tolerance with a reinforcement learning approach.

This paper has a novel contribution, outperforms the baseline (CCNF: CNF (Chen et al.) + GMM / auxiliary classifier (Kingma & Dhariwal, 2018)) on several evaluation metrics. I agree to accept this paper, but I vote for ‘weak accept’ because of the following weaknesses:

1. The (three) contributions described by the authors seem to be somewhat exaggerated.
- For the second contribution, the authors’ way is quite similar to Wang et al. (SkipNet). For both studies, the purpose is the efficiency of certain neural architectures and training scheme is hybrid reinforcement learning with REINFORCE (but with different reward design).
- For the last contribution, I think it is a minor contribution comparing to two first bullets and overlapping with the second one.

2. There is a lack of explanation to support the efficiency of the proposed model.
- The authors claim that InfoCNF needs fewer parameters than the baseline. Then, why didn't you show the actual number of parameters? The only line that I found was “... InfoCNF requires 4% less parameters than CCNF. (Section 2)”.
- Also, it would be better if there were a direct computation and comparison between the size of InfoCNF and CCNF.
- Finally, is there any constraint on the length of merged latent code z? Since InfoCNF is also an invertible model, it should have the same size as the input, but I cannot find descriptions about it.

3. It is skeptical that automatic tuning of error tolerance has achieved its original purpose.
- For me, it is questionable whether automatic tuning has achieved its original purpose: reducing NFEs for better speed (and performance).
- In figure 3c and 3f, we can find NFE of InfoCNF learned tolerances eventually exceeds the NFE of InfoCNF fixed tolerance. It looks like learning tolerance increases NFE in large epochs, and the timing seems to depend on the batch size. If so, the batch size is a really important hyper-parameter in this framework, how can we determine the appropriate size?
- In section 4.4 (Automatic Tuning vs. Manual Tuning), the authors state that “our automatic approach learns the tolerances which outperform the manually-tuned ones in both classification and density estimation while being only slightly worse in term of NFEs.”. But as abstract says, is reducing NFEs the original goal of automatic tuning?
- Lastly, how can we know/confirm “our automatic approach via reinforcement learning requires much less time and computational budget”. I cannot see any explanation about this claim.

4. Minor things.
- What is the stage stated in Figure 1?
- The abbreviation of the CCNF first appears in section 1 but its full name first appears in section 2.2.
- In figure 3a and 3b (and table 1), why is test NLLs of CCNF lower than InfoCNF’s where test error of CCNF is larger than InfoCNF’s with high margin? Is there any possible explanation?

**Experience Assessment:**

I do not know much about this area.

**Review Assessment: Checking Correctness Of Derivations And Theory:**

I assessed the sensibility of the derivations and theory.

**Review Assessment: Checking Correctness Of Experiments:**

I carefully checked the experiments.

**Review Assessment: Thoroughness In Paper Reading:**

I read the paper at least twice and used my best judgement in assessing the paper.

---

> ### Author Response · Authors · 2019-11-14
> **Automatic Tuning vs. Tuning vs. No Tuning**
>
> In figure 3c, the number of function evaluations (NFEs) in InfoCNF with learned tolerances only exceeds the NFEs in InfoCNF with fixed tolerances (no tuning) at the end of the training. Overall, the training of InfoCNF with learned tolerances is still much faster than the training of the same model with fixed tolerances, as well as the training of the baseline CCNF model (775.98 NFEs/epoch vs. 984.92 NFEs/epoch vs. 924.12 NFEs/epoch). Please refer to our reply to Reviewer #3: "Using Reinforcement Learning to Learn the Error Tolerance is Beneficial and More Time Efficient" for the training time per epoch of these models.
>
> In figure 3f, as mentioned in Section 4.4 - Automatic Tuning vs. Manual Tuning, we carefully designed the fixed tolerances (manual tuning) to make sure that InfoCNF could be trained and yielded good results when training with large batches. However, this process took a lot of time since we needed to do hyper-parameter search for the error tolerance among the values 1e-5, 1e-4, 1e-3 for each solver in the model. The message we would like to convey in Figure 3d, e, f is that by using gating networks to automatically learn the error tolerances, we can achieve results which are on a par with the manual tuning approach via hyper-parameter search. The same results hold for small batch training as shown in Section F of our Appendix. Figure 10 in this section compares the test error, the NLL, and the NFEs of InfoCNF with learned tolerances, manually tuned tolerances, and fixed tolerances (no tuning) with those of the baseline CCNF for small-batch training of batch size 900.
>
> We have also conducted additional trainings on CIFAR10 using different batch sizes: 900, 1800, 2400, 3200, 5400, and 8000. We observe that InfoCNF with learned tolerances consistently yields smaller NFEs than InfoCNF with fixed tolerances (no tuning) and the baseline CCNF while yielding similar or only slightly higher NFEs than InfoCNF with manually tuned error tolerances. Furthermore, we notice that when training InfoCNF with learned error tolerances, the larger the batch size, the smaller the NFEs is, but the worse the NLLs and test errors are. However, compared with the small batch size of 900, there are some intermediate batch size such as 1800 that yields only slightly worse NLLs (3.619 vs 3.568) and test errors (21.45% vs. 20.99%) but results in significantly faster training with much smaller NFEs (618  vs. 755 per epoch).

---

> ### Author Response · Authors · 2019-11-14
> **Replies to Other Comments from Reviewer #4**
>
> 1) Our 2nd and 3rd Contribution:
>
> Our learning error tolerance approach is inspired by the SkipNet in Wang et. al. To the best of our knowledge, we are the first one to employ this approach to learn the parameters of an ODE solver. We agree with the reviewer that the experiments done in the last contribution is to validate the advantages of the first and second contribution.
>
> 2) Number of Parameters in Our Models:
>
> The number of parameters in CCNF is 1,469,494 while the number of parameters in InfoCNF without and with gating networks are 1,414,198 and 1,450,732 respectively.
>
> 3) Constraint on the Length of Merged Latent Code z:
>
> The latent code z must be of the same size as the input image to guarantee invertibility. We then split z into the supervised code z_y and the unsupervised code z_u. We only use the supervised code z_y for conditioning and classification, thereby reducing the size of the Gaussian mixture model used for conditioning and the fully-connected layer used for classification.
>
> 4) Why is Test NLLs of CCNF Lower than InfoCNF’s where Test Error of CCNF is Larger than InfoCNF’s with High Margin?
>
> There is a tradeoff between achieving good classification accuracy and obtaining lower NLLs.  We show that InfoCNF can achieve similar or only slightly worse NLLs compared to the baseline CCNF while significantly improves the classification accuracy, which is the advantage of our model.
>
> 5) Stage Stated in Figure 1:
>
> A stage in Figure 1 is a processing block in the multiscale architecture used in [1]. Each processing block contains multiple ODE units, and the dimensionality of the activations is preserved within each block. The multiscale architecture reduces the dimensionality of the activations after each processing block while preserving invertibility. This architecture was first introduced in [2] and has been widely used as the base architecture for invertible models.
>
> References:
>
> [1] W. Grathwohl, R. T. Chen, J. Betterncourt, I. Sutskever, and D. Duvenaud. Ffjord: Free-form continuous dy358 namics for scalable reversible generative models. In International Conference on Learning Representations, 359 2018.
>
> [2] L. Dinh, J. Sohl-Dickstein, and S. Bengio. Density estimation using real nvp. In International Conference 348 on Learning Representations, 2017.

---

### Official Review · AnonReviewer3 · 2019-11-04
**Official Blind Review #3**

**Rating:** 1

**Review:**

This paper proposes a conditioning approach for CNF and explore speed-up by tuning error tolerance of the ODE solvers.

Overall, this is a mediocre paper which directly use a similar structure introduced in InfoGAN but absolutely lose beautiful insights in the construction of the loss function in InfoGAN, i.e., the mutual information between the generated image and codes. In other way, this paper is just an incremental paper with even less insights than the seminal paper.
At least in the loss function Eq. (4), I didn't see any mutual information regularization used here. Instead, the authors use a GMM but I am totally not sure why a GMM is better than the mutual information regularization. At the same time, in equation (4), I didn't see the specific definition of L_xent and L_NLL and thus I am not even able to verify that the use of the loss function is correct. For the current version, I am not be able to gain any insight from the loss function. It seems to be an ensemble of several existing works and definitely not innovative.

For the tuning error of the ODE solvers, I didn't even see the problem statement.  Is it possible to make the problem more clear? It seems that the first time when the authors mentioned error tolerance is in contribution 2, but I didn't see the definition of the error tolerance. I am not sure whether it is a good idea to introduce two problems in one paper. At the same time, I do not know why the problem should be formulated in the form of the reinforcement learning problem. I didn't see any advantage.  Intuitively, learning a generative model can be time consuming  and solving a reinforcement learning problem is also hard. I do not understand why combining them together would be beneficial and even time efficient? For me, it seems that authors are just trying to make the problem unnecessarily more complicated and thus they can use fancy tools to solve it.



**Experience Assessment:**

I have read many papers in this area.

**Review Assessment: Checking Correctness Of Derivations And Theory:**

N/A

**Review Assessment: Checking Correctness Of Experiments:**

I carefully checked the experiments.

**Review Assessment: Thoroughness In Paper Reading:**

I read the paper thoroughly.

---

> ### Author Response · Authors · 2019-11-14
> **InfoCNF Preserves the Beauty of InfoGAN**
>
> InfoCNF does not lose the beauty of InfoGAN. We will briefly summarize the beauty of InfoGAN below and then explain why our InfoCNF preserves such beauty, but is more efficient.
>
> The beauty of InfoGAN lies in the way the mutual information I(c; G(z,c)) is approximated by a variational lower bound. This lower bound is computed via an “auxiliary” distribution Q(c|x), which approximates the real posterior P(c|x). Q(c|x) is chosen to be a neural network. A re-parameterization trick is then employed to allow sampling from the prior instead of the unknown posterior. In the context of GANs, this process is implemented as follows: 1) Sample a value of the latent code c from the prior distribution; 2) Sample the noise z, for example, from a uniform distribution; 3) Generate x = G(c,z); 4) Calculate Q(c|x=G(c,z)).
>
> The beauty of InfoCNF lies in its invertibility. Since InfoCNF is an invertible model, it consists of a forward model G  - which generates image x given the latent code c and the noise z - and an inverse model G^{-1} - which computes c and z given the input image x. G and G^{-1} share weights and are invertible to each other. Thus, for InfoCNF, G(c,z) in 4) above reconstructs the exact input image x. Maximizing log (Q(c|x=G(c,z))) is then equivalent to maximizing the cross-entropy loss between the estimated label and the ground truth label, which is the loss L_{Xent}(y_hat, y) in Equation 4 of our paper. Note that the latent code c in InfoCNF is computed by the inverse model G^{-1} which shares weights with the forward model G, thereby eliminating the need of using an additional “auxiliary” network Q.
>
> We also want to note that re-parameterization via GMM is a popular method to condition a flow-based model as used in [1].
>
> Reference:
> [1] Durk P Kingma and Prafulla Dhariwal. Glow: Generative flow with invertible 1x1 convolutions. In Advances in Neural Information Processing Systems, pp. 10215–10224, 2018.

---

> ### Author Response · Authors · 2019-11-14
> **Using Reinforcement Learning to Learn the Error Tolerance is Beneficial and More Time Efficient**
>
> First, learning the error tolerances in the solvers helps reduce the number of function evaluations in infoCNF, which makes training faster. Even though using gating networks to compute the error tolerances introduces more parameters to learn, it turns out that overall infoCNF with gating networks can be trained faster than infoCNF without gating networks. This is because the gating networks we use only have one convolutional layer with a small number of parameters. We include the training time for the infoCNF with and without gating networks, as well as the training time for the baseline CCNF, averaged over 3 trials below. The training time is also averaged over epochs.
>
> CCNF:                                                                    1362 seconds/epoch
> InfoCNF without learning error tolerance:    1440 seconds/epoch
> InfoCNF with learning error tolerance:          1270 seconds/epoch
>
> Second, since learning error tolerances of the ODE solvers is a non-differentiable optimization problem, we need to employ REINFORCE to update parameters of the gating networks.

---

> ### Author Response · Authors · 2019-11-14
> **Replies to Other Comments from Review #3**
>
> 1) Partitioning the Latent Code and Learning the Error Tolerances are Complementary to Each Other
>
> Both approaches try to speed up and increase the efficiency of the Continuous Normalizing Flow (CNF). Since the partitioning yields more number of function evaluations than the baseline in small-batch training, we develop the error tolerance learning via REINFORCE to overcome this drawback.
>
> 2) Definition of Error Tolerance, Problem Statement for Tuning the Error Tolerances of the ODE Solvers, and Definition of L_xent and L_NLL
>
> ODE solvers can guarantee that the estimated solution is within a given error tolerance of the true solution. Increasing this tolerance enhances the precision of the solution but results in more iterations by the solver, which leads to higher NFEs and longer training time. However, when training a neural network, it might not be necessary to achieve high-precision activations, i.e. the solution of the corresponding ODE, at each layer. Some noise in the activations can help improve the generalization and robustness of the network [1,2,3,4]. With carefully selected error tolerances, InfoCNF can gain higher speed and better performance.
>
> We have updated our paper to include the definition of error tolerance and problem statement for tuning them in the Contribution 2 in the introduction of our paper. We have also provided the definition for L_xent and L_NLL in equation (4) and (5) at the end of section 2.2.
>
> References
>
> [1] Caglar Gulcehre, Marcin Moczulski, Misha Denil, and Yoshua Bengio. Noisy activation functions. In International conference on machine learning, pp. 3059–3068, 2016.
> [2] Yoshua Bengio, Nicholas Léonard, and Aaron Courville. Estimating or propagating gradients through stochastic
> neurons for conditional computation. arXiv preprint arXiv:1308.3432, 2013.
> [3] Vinod Nair and Geoffrey E Hinton. Rectified linear units improve restricted boltzmann machines. In Proceedings
> of the 27th international conference on machine learning (ICML-10), pp. 807–814, 2010.
> [4] Bao Wang, Binjie Yuan, Zuoqiang Shi, and Stanley J Osher. Enresnet: Resnet ensemble via the feynman-kac formalism. arXiv preprint arXiv:1811.10745, 2018a.

---

### Decision · Program_Chairs · 2019-12-19

**Decision:**

Reject

**Comment:**

This paper presents a conditional CNF based on the InfoGAN structure to improve ODE solvers. Reviewers appreciate that the approach shows improved performances over the baseline models.

Reviewers all note, however, that this paper is weak in clearly defining the problem and explaining the approach and the results. While the authors have addressed some of the reviewers concerns through their rebuttal, reviewers still remain concerned about the clarity of the paper.

I thank the authors for submitting to ICLR and hope to see a revised paper at a future venue.